# The Functional Significance of High Cysteine Content in Eye Lens γ-Crystallins

**DOI:** 10.3390/biom14050594

**Published:** 2024-05-17

**Authors:** Eugene Serebryany, Rachel W. Martin, Gemma R. Takahashi

**Affiliations:** 1Department of Physiology & Biophysics, Stony Brook University, SUNY, Stony Brook, NY 11794, USA; 2Laufer Center for Physical & Quantitative Biology, Stony Brook University, SUNY, Stony Brook, NY 11794, USA; 3Department of Chemistry, UCI Irvine, Irvine, CA 92697-2025, USA; 4Department of Molecular Biology & Biochemistry, UCI Irvine, Irvine, CA 92697-3900, USA

**Keywords:** eye lens, cataract, crystallin, cysteine, methionine, disulfide, protein misfolding, protein aggregation, refractive index, protein evolution

## Abstract

Cataract disease is strongly associated with progressively accumulating oxidative damage to the extremely long-lived crystallin proteins of the lens. Cysteine oxidation affects crystallin folding, interactions, and light-scattering aggregation especially strongly due to the formation of disulfide bridges. Minimizing crystallin aggregation is crucial for lifelong lens transparency, so one might expect the ubiquitous lens crystallin superfamilies (α and βγ) to contain little cysteine. Yet, the Cys content of γ-crystallins is well above the average for human proteins. We review literature relevant to this longstanding puzzle and take advantage of expanding genomic databases and improved machine learning tools for protein structure prediction to investigate it further. We observe remarkably low Cys conservation in the βγ-crystallin superfamily; however, in γ-crystallin, the spatial positioning of Cys residues is clearly fine-tuned by evolution. We propose that the requirements of long-term lens transparency and high lens optical power impose competing evolutionary pressures on lens βγ-crystallins, leading to distinct adaptations: high Cys content in γ-crystallins but low in βB-crystallins. Aquatic species need more powerful lenses than terrestrial ones, which explains the high methionine content of many fish γ- (and even β-) crystallins. Finally, we discuss synergies between sulfur-containing and aromatic residues in crystallins and suggest future experimental directions.

## 1. Introduction

Cysteine is the rarest canonical amino acid in proteins; its frequency increases with organismal complexity, from about 0.5–1% in prokaryotes to 2% in mammals, all of which are low compared to the expected frequency of about 3.3% for an amino acid with two codons [1,2]. Thus, it appears that cysteine is frequently selected against by evolution, potentially because of its reactivity. It is uniquely capable of readily reversible redox reactions: the formation and breakage of disulfide bonds, along with several other types of oxidative modifications. It is a plausible hypothesis that the importance and sensitivity of redox chemistry in an organism’s life cycle plays an increasingly important role roughly in proportion to the percentage of Cys residues in its proteome. For example, *E. coli* can tolerate wide variations in its cytoplasmic redox potential—a feature that has been harnessed to great advantage for the heterologous expression of disulfide-containing proteins [3,4]. By contrast, alterations of the redox potential setpoint in human cells, in either direction, are strongly associated with the development of cancer [5,6,7]. In general, accumulating evidence links proteomic redox balance to the aging process [8,9,10].

Cataract, caused by light-scattering aggregation of natively highly soluble eye lens proteins, crystallins, is one of the most common diseases of aging, afflicting the majority of Americans over 65 (self-reported diagnoses [11]) and accounting for 50% of all cases of blindness in low- and middle-income countries [12]. It is a striking example of the nexus of molecular biophysics and biochemistry with organismal aging. Most cataract occurs in the central (or nuclear) region of the eye lens, but peripheral (cortical) cataract is also common [13]. The proteome of terminally differentiated lens fiber cells is dominated by highly concentrated α-, β-, and γ-crystallins—soluble monomeric or oligomeric proteins that are never replaced [14,15]. There are two superfamilies [16] of lens crystallins in humans: (1) the α-crystallins are small heat-shock proteins that act as molecular chaperones [17,18], and (2) the β- and γ-crystallins, which are the primary structural and refractive proteins [19]. The α and βγ crystallin families are present in all vertebrates; additional taxon-specific lens crystallin families exist, such as the δ-crystallins of many avians and reptiles [20,21]. Our focus here is on the βγ-crystallins, which share a common double-Greek key fold and a two-domain structure held together by an interdomain interface [22,23,24], and on the distinctions between the β- and γ-crystallin families. The β-crystallins are found in domain-swapped [25] or face-en-face dimers [26] and can form higher-order oligomers [27] in contrast to the monomeric γ-crystallins [28]. In addition to these crystallins, which are common to all vertebrates, many organisms (although not humans) also have taxon-specific crystallins recruited from diverse small enzyme families [29].

The cytoplasmic redox potential of the extremely long-lived lens fiber cells drifts predictably with age toward increasingly oxidizing values, leading to the formation of many disulfide bonds in the crystallin proteome [30,31,32,33,34,35]. The severity of age-onset cataract depends on the extent of crystallin disulfide bonding and on which specific Cys residues are involved [31,36,37]. In vitro experiments by ourselves and others have demonstrated clear evidence that γ-crystallins, in particular, can act as oxidoreductases, exchanging disulfides among themselves, forming a de facto protein-based redox buffer to compensate for the depletion of the glutathione redox buffer in the aging lens [38,39].

While some disulfide bonds can form in native or native-like conformations of the γ-crystallins, others trap non-native, aggregation-prone conformations [38,40]. Exposure of buried Cys residues can also create new binding sites for transition metals, which accumulate in the lens with age and particularly with cataractogenesis [41,42,43,44,45]. In turn, there is evidence of extensive regulation of the reactivity of Cys residues in the aging lens, ranging from partial methylation of exposed Cys residues [46] to the formation of irreversible oxidation products at “sacrificial” Cys residues [47]. Moreover, the lens metabolome appears to have evolved to suppress the conformational dynamics of the γ-crystallins that would otherwise lead to the formation of non-native disulfide bonds favoring misfolded, aggregation-prone γ-crystallin conformations [40].

In light of the evolution of the βγ-crystallin family, however, all the above observations present a surprising paradox. The γ-crystallins and β-crystallins are closely related both evolutionarily and structurally, and they coexist in the cytoplasm of the same lens fiber cells at high concentrations. Yet, neither the number, nor the positions of Cys residues is conserved within this protein family: as shown in Appendix A, the most conserved residue types are Gly, Ser, and aromatics—all crucial for the highly compact double-Greek key fold—but not Cys. Although γ-crystallins are universally cysteine-rich, βA-crystallins have relatively typical content for proteins of their length, while βB-crystallins are cysteine poor (see Table 1 and Appendix A). As a result, multiple sequence alignment using the DeepMSA2 algorithm [48,49] shows quite low conservation of Cys residues in the βγ-crystallins overall, compared to the high conservation of Ser, Gly, and aromatic residues important for proper folding of the topologically complex Greek key motifs (Figure 1). Given the highly deleterious consequences of disulfide-trapped misfolding and subsequent light-scattering aggregation leading to cataract, why did the γ-crystallins not evolve to be largely or entirely Cys free, as the βB-crystallins have done? We propose two potential hypotheses: (1) Cys is selected because it increases refractivity, even at the cost of deleterious aggregation later in life; and (2) the spatial positioning of Cys residues in the crystallin structure is more important than the total number of Cys residues for maintaining the monomeric state of γ-crystallins.

A common misconception regarding diseases of aging is that human life expectancy in the era before modern medicine was much shorter than today and did not extend much past peak reproductive years; therefore, evolution has not had a chance to “solve” protein misfolding diseases or diseases of aging in general. This persistent belief relies on incorrect assumptions. First, average life expectancy in premodern societies is an irrelevant metric of adult human lifespan because the average is heavily skewed by high infant and child mortality. It is well documented, both in ancient sources and by observations of modern hunter-gatherer societies and societies with little or no modern medical care, that humans who survived childhood and had proper nutrition frequently lived to their 70s and even beyond [50,51]. Second, grandparents contribute substantially to the evolutionary fitness of their grandchildren [52,53]. For these reasons, our species has likely faced strong evolutionary pressure to maintain clarity of vision well into old age. In this way, cataract disease is akin to, e.g., Huntington’s disease: the protein misfolding is nearly inevitable from the physico-chemical standpoint, but the proteins and their environment have both been tuned by evolution to delay such an outcome for a lifetime [54]. Furthermore, other long-lived vertebrates are reproductively active until late in life and lack the human social structures that can allow for survival of blind individuals, so we should not consider old-age lens transparency as beyond the reach of evolution. As pointed out already by W. D. Hamilton, evolution tends to select for early-life fitness at the cost of late-life senescence, yet human evolution has clearly selected for a long post-reproductive lifespan, too [55].

Mere genetic drift clearly does not provide the explanation, either. Evolutionary pressure on γ-crystallin sequences has been high, as evidenced by their highly unusual amino acid composition. For example, the ratio of Lys to Arg residues in human proteins is typically ~1:1 [56], but this ratio within the abundant human γ-crystallins (γC, γD, and γS) is ~1:4. Meanwhile, the ratio of Ala to Cys in human proteins is ~3:1 [56], but in the γ-crystallins, it is ~2:3. Fish γ-crystallins have even more extreme compositions; several crystallins from the Antarctic toothfish (*Dissostichus mawsoni*) contain no alanine and are instead highly enriched in highly polarizable residues, including Phe, Tyr, Arg, and Met [57]. This leads to unusual biophysical properties, including differential resistance to thermal and chemical denaturation [58], as well as resistance to cold cataract far below temperatures encountered by the fish [59]. One likely biophysical explanation for this lies with the differential propensity of Lys-enriched vs. Arg-enriched γ-crystallin variants to undergo liquid–liquid phase separation at the very high (100 s of mg/mL) protein concentrations involved [59,60].

Why are Cys residues within the eye lens proteome so concentrated in the γ-crystallins and exiguous in many of the closely related β-crystallins? Here we discuss the γ-crystallins’ high Cys content in relation to two major evolutionary requirements for eye lens function: (1) high long-term transparency, i.e., minimization of light-scattering aggregation, which is achieved via thermodynamic and kinetic stability, plus oxidoreductase activity; and (2) high optical power, i.e., maximization of the refractive index increment of the protein to facilitate the focusing of light, which is achieved via highly unusual amino acid compositions, plus tertiary structural features. We use multiple sequence alignments and phylogenetic clustering at both the amino acid and the nucleotide levels to draw evolutionary relationships among lens crystallins. We use the most successful current protein structure prediction algorithm, D-I-TASSER [61], to compile a dataset of predicted γ-crystallin structures from phylogenetically representative species. This combination of methods reveals potential evolutionary signatures of thiol/disulfide chemistry in the eye lens. We hypothesize that the requirements of high optical power (refractive index increment) and high long-term transparency (aggregation avoidance) are sometimes in tension with each other and may have pushed the β and γ families of lens crystallin to opposite extremes of Cys content.

## 2. Molecular Etiology of Cataract and the Paradox of High Cys Content

### 2.1. Thermodynamic and Kinetic Stability of βγ-Crystallins

Human γ-crystallins are among the most thermodynamically stable in the body [62]. The most stable of these is γD-crystallin: a well-structured, monomeric, two-domain, double-Greek-key, predominantly β-sheet protein that cannot be unfolded by a saturated solution of urea at neutral pH [63]. This protein is also thermostable up to ~80 °C [37,64]. Other human γ-crystallins are also highly thermodynamically stable proteins, and they also show exceptional kinetic stability, with unfolding half-lives of minutes to hours even in 4 M guanidinium [65,66]. Notably, γ-crystallins from mice appear to be less thermodynamically stable than their human counterparts [67]. How crystallin stability varies among animal species has not yet been systematically investigated.

By contrast, the β-crystallins, which share the same fold and high sequence identity to the γ-crystallins, are more thermodynamically labile. Although they are still highly stable compared to the average human protein, members of this family are urea denaturable [68,69] and have melting temperatures some 10–20 °C below the γ-crystallins [70]. Unlike the monomeric γ-crystallins, the β-crystallins are also known to form a variety of oligomeric complexes [68,71]. Nevertheless, β-crystallins are more highly abundant than γ-crystallins in the human lens [72]. One important reason may be their lower redox sensitivity: as noted in the Introduction above, cysteine residues are the Achilles’ heel of γ-crystallins; although they are likely beneficial under the reducing conditions of young lens cytoplasm, later in life they acquire disulfide bonds that distort the native structure and promote aberrant oligomerization in the more oxidizing cytoplasm of aged lenses [32,36,73]. The βB-crystallins, being exiguous in Cys, are accordingly less susceptible to the effects of changing thiol chemistry with age.

### 2.2. Aggregation Propensity of γ-Crystallins

Cataract is increased turbidity (visible light scattering) of the lens [14,74]. It is typically a disease of aging, although other factors, such as eye trauma, can promote cataract development [75,76]. Once the size of protein aggregates becomes comparable to the wavelength of visible light, these particles scatter increasing amounts of visible light photons before they can reach the retina. The result is the blurring of vision and a red shift in the perceived color palette [77], because the shorter blue wavelengths of light are scattered more strongly by small aggregate particles than the longer red wavelengths. However, due to the enormous challenge of in vivo structural biology, particularly of protein aggregates, there remains some controversy over the exact nature of protein aggregation during typical age-onset cataract disease, as discussed below. We will briefly summarize available evidence from in vitro, in vivo, and ex vivo studies, although a comprehensive overview of the crystallin aggregation literature is beyond the scope of the present focused review.

Aggregation of γ-crystallins has been studied in vitro for a half-century but continues to reveal new and surprising biophysical phenomena. Lens crystallins natively exist at high concentrations and have clearly evolved to push the biophysical envelope on solubility, stability, and a plethora of fascinating mechanisms of aggregation avoidance. Yet, they are not invulnerable. In fact, in vitro biophysical and biochemical research on γ-crystallins has revealed every conceivable mode of aggregation in these highly natively stable proteins. Rare cataract-associated variants of γ-crystallins have been shown to aggregate via crystallization [78], amyloid formation [79], amorphous aggregation without major conformational change [80], and redox-driven aggregation from partially unfolded states via a domain-swap-like mechanism [37,81,82,83]. They can also form liquid droplets in their folded states, thus undergoing liquid–liquid phase separation [59,60], or form aggregates wherein the monomers are either bridged or misfolded (or both) by transition metal cations [41,84,85]. Predominantly amorphous (and some amyloid) aggregation can be induced by exposure to UV light [47,86,87], which is a physiologically relevant stressor for the lenses of terrestrial animals.

In vivo and ex vivo studies of crystallin aggregation likewise have a long history, and it has been a tremendous yet motivating challenge to extend the tools of structural biology and molecular biophysics from the test-tube to the unique (and uniquely accessible) environment of the lens. For example, the transparency of the lens has enabled pioneering work on in vivo dynamic light scattering for improved diagnostics of cataract, and potentially even Alzheimer’s progression [88,89,90], as well as the first in vivo application of two-dimensional infrared spectroscopy (2D-IR) that is highly attuned to detecting amyloid-like β-structures [91], while electron microscopy of lens slices revealed increased cytoplasmic texturing that could account for increased light scattering without fibrils [92]. The phenomenon of cold cataract was the first to be extensively researched, and indeed temperature-driven “cold precipitation” (later understood to arise via protein condensate formation) in the lens crystallins was already reported in 1964 [93]. Crystals extracted directly from the lenses of patients carrying the rare point mutations of human γD-crystallin that lead to crystal cataracts yielded some of the highest-resolution protein crystal structures at the time [78]. Other research has emphasized that proteomes evolve at the edge of insolubility [94,95], and broad insolubilization may sometimes occur in the lens [96].

However, crystallin aggregates extracted from age-onset cataract lenses were neither crystalline nor temperature reversible; an early demonstration that the combination of strong denaturant and reducing agent could disassemble these aggregates establishing age-onset cataract as a protein misfolding disease with an important component of redox chemistry [31]. Since then, proteomic studies have revealed a large and growing number of post-translational modifications in lens proteins as a function of age, including deamidation, Asp isomerization, oxidation of Met, Cys, and Trp, Cys methylation, and Lys succinylation [46,97,98,99]. Notably, the formation of a diffusion barrier roughly coincident with the boundary between the lens nucleus and the lens cortex has been observed, and it correlates with the drift in the redox potential of lens fiber cell cytoplasm to increasingly oxidizing levels beginning in midlife [34,100,101,102]. For the Cys-rich γ-crystallins, but also for those α- and β-crystallins that contain at least two Cys residues, formation of intramolecular disulfide bonds correlates strongly with cataract progression [36]. Intermolecular disulfides have also been observed in the lens [73], and both intra- and intermolecular disulfides have been shown to promote aggregation [37,38,40,81,103,104] or inhibit chaperone function [105] in vitro. Sustained research efforts have therefore focused on the chemistry of crystallin Cys residues as an important contributor to the etiology and progression of age-onset cataract [32,106,107,108].

## 3. Evolutionary and Structural Analysis of Chordate βγ-Crystallins

To better understand the evolutionary origin and importance of high Cys content in γ-crystallins, we compared γ-crystallins from representative chordates to one another and, as a group, to βA- and βB-crystallins from the same organisms. For each organism, we selected every protein annotated in UniProt as a γ-crystallin (or as βA- and βB-crystallin, respectively) and containing either two (for the single-domain βγ-crystallin from *Ciona intestinalis*) or four (all other organisms) complete Greek key domains. A small number of truncated crystallins were excluded, as were some homologs of absent-in-melanoma-1 (AIM-1), which has several βγ-crystallin domains but is not a lens protein [109].

Evolutionary comparisons allow us to test several hypotheses as to why lens γ-crystallins are cysteine rich. First, if Cys residues at specific positions are important in stabilizing the double-Greek key domains, then those Cys positions should be conserved among these and homologous proteins. Second, if sulfur-containing residues are favored in lens crystallins due to their high polarizability and consequently high refractive index, then the lens crystallins of aquatic animals should be especially Cys rich (as well as Met-rich) compared to land-dwelling animals who are able to derive significant optical power from the cornea. Third, if avoidance of disulfide-driven aggregation or changes in short-range packing are a major evolutionary pressure, then the natively monomeric γ-crystallins should have no more than one solvent-exposed Cys-rich site per molecule: enough to facilitate redox chemistry but not enough to undergo disulfide-driven “daisy-chain” type aggregation from the native state. All these hypotheses can be tested by a combination of evolutionary and structural analysis that is now possible thanks to the rapid growth in available genomic data on the one hand and machine-learning driven protein structural prediction methods on the other. We note, however, that these hypotheses will ultimately require direct experimental testing, some of which is already underway in our labs.

### 3.1. Lack of Cys Conservation in βγ-Crystallins

We used the DeepMSA2 (https://zhanggroup.org/DeepMSA/) algorithm [49] to generate residue conservation logos across the sequence, for four representative crystallins: the human γD- and γS-crystallins, a fish γM-crystallin, and an ancestral-like βγ-crystallin from tunicates (Appendix A). By default, this tool measures residue conservation across all protein-coding sequences in metagenomic databases that can be successfully aligned to the sequence of interest (typically thousands or tens of thousands of sequences). Considering that γ-crystallins, and even many β-crystallins, are enriched in cysteine, the result is surprising: there is very low conservation of Cys residues. Instead, Ser, Gly, and aromatic residues, all of which are likely critical for the proper folding and stability of crystallin domains, are by far the most conserved residues. Several charged residues are also conserved. While this result may be skewed by a high abundance of microbial βγ-crystallins in the metagenomic databases, it indicates at least that crystallin domains generally do not require Cys residues.

Why, then, are Cys residues so abundant in the γ-crystallins? To gain a more fine-grained understanding of this question, we carried out phylogenetic clustering of γ-crystallin sequences and structural analysis of Cys and Met positions and solvent exposure. Here we primarily focus on γ-crystallins because these are the most enriched in Cys residues, and also the most refractive proteins, in the mammalian lens [110]; the abundance of Cys and Met in β-crystallins is provided for comparison in Appendix A.

### 3.2. Phylogenetic Clustering of γ-Crystallins

The protein set chosen for this discussion consists of all γ-crystallin isoforms reported in UniProt [111] for the following vertebrates: *Petromyzon marinus* (sea lamprey), *Latimeria chalumnae* (coelacanth), *Chiloscyllium indicum* (slender bamboo shark), *Danio rerio* (zebrafish), *Xenopus laevis* (African clawed frog), *Ornithorhynchus anatinus* (platypus), *Macropus fuliginosus* (western gray kangaroo), *Mus musculus* (mouse), *Bos taurus* (cow), *Homo sapiens* (human), *Anolis carolinensis* (green anole), *Alligator mississippiensis* (American alligator), and *Gallus gallus* (chicken). Additional βγ-crystallins from the chordates *Eptatretus burgeri* (inshore hagfish), *Ciona intestinalis* (tunicate) and *Branchiostoma floridae* (lancelet) were included for comparison. The same organisms’ β-crystallin sequences were likewise retrieved from UniProt for comparison.

Protein sequences from the crystallins were aligned using Clustal Omega v1.2.4 [112] and custom scripts (github.com/grtakaha/protein_alignment_tool) written in and tested on Python v3.7.4 [113]. Sequence alignments for the γ- and β-crystallin are shown in Appendix A, respectively. Conserved residues from large numbers of related sequences were identified using DeepMSA2 (https://zhanggroup.org/DeepMSA/) [48,49]. Residue conservation was visualized using WebLogo (https://weblogo.threeplusone.com/) [114].

Nucleotide sequences (exons only) were retrieved using the following methods depending on availability of information in UniProt [111].

Method 1: extracted using exon coordinates and an NCBI nucleotide ID (Nucleotide database) [115] from the “Genomic Coordinates” tab of the given UniProt entry.

Method 2: identified within an NCBI reference sequence entry (Nucleotide database) [115] from the “Sequence databases” section of the main page of the given UniProt entry.

Method 3: retrieved from an Ensembl [116] ID (gene or transcript), found in the “Genome annotation databases” section of the main page of the given UniProt entry; only “cds:protein coding” sequences were used.

Method 4: retrieved from a “PROTEIN SEQUENCE” EMBL [117] link to the European Nucleotide Archive (ENA) [118] found in the “Sequence databases” section of the main page of the given UniProt entry.

A multiple sequence alignment of all γ-crystallin protein sequences was further generated using Clustal Omega v1.2.4 [112], and the results were used to generate a tree based on protein sequence identity. Specifically, the percent identity matrix from this alignment was used to make a dissimilarity matrix (100—percent identity), which was then used as input for hierarchical clustering via the ward.D2 method of the hclust function in the base R v4.1.1 Stats package [119]. Clusters were visualized in a circular dendrogram using circlize v0.4.15 [120] and dendextend v1.17.1 [121] (Figure 1). This type of tree does not show any information about evolutionary relationships: it does not account for synonymous mutations at the nucleic acid level or convergent evolution toward similar sequences. The crystallin sequences cluster into four groups. The most basal cluster (Cluster 1) contains crystallins from the tunicate, lancelet, and hagfish, consistent with the positioning of these organisms near the base of the chordate/vertebrate family tree. The next group to split (Cluster 2) contains γS-crystallins, which are found in most vertebrates, including hagfish, bony fish, lobe-finned fish, mammals, reptiles, and birds. From there, the remaining crystallins are divided into two groups, one containing mammalian γD-crystallins as well as similar proteins from other organisms (Cluster 3), and the other (Cluster 4) containing fish γM-crystallins. In this set, most of the γM-crystallins come from the zebrafish, *Danio rerio*, with one sequence belonging to the slender bamboo shark.

For the γ-crystallins listed in Figure 2, nucleotide sequences were aligned by codons using MUSCLE [122] through the MEGA X v10.1.8 interface [123] (Gap Open = −2.90, Gap Extend = 0.00, Hydrophobicity Multiplier = 1.20, Max Iterations = 16, Genetic Code = Standard, Cluster Methods = UPGMA, Min Diag Length (Lambda) = 24). This alignment was used to generate a tree (protein coding option) in MEGA X v10.1.8 [123] using the Neighbor-Joining method [124] with a bootstrap test [125] of 1000 replicates. Distances (base substitutions per site) were calculated using the Maximum Composite Likelihood method [126]. Ambiguous comparisons were handled by pairwise deletion.

Surprisingly, many proteins that were previously annotated as γM or γS cluster with the γD-crystallins, including four proteins from the zebrafish and three from the slender bamboo shark. The zebrafish proteins most strongly resemble orthologs from either the coelacanth or the alligator. The mammalian γD-crystallins are strongly similar to one another but are otherwise unremarkable with respect to the other proteins in Cluster 4; they are located in the middle of the larger cluster. Fish are a polyphyletic group, with three distinct lineages, the chondrichthyans (including sharks and rays), the teleosts such as the zebrafish, and the lobe-finned fish like the coelacanth, which are more closely related to the tetrapods [127,128]. The *Xenopus laevis* γ-crystallins cluster with this group as well, raising the possibility that the hypothesis of Bloemendal et al. that these proteins arose from a distinct gene duplication relative to γS may also apply to other γD-like crystallins [23].

A phylogenetic tree based on the nucleic acid sequences coding for these same proteins is shown in Figure 2. At the nucleic acid level, these sequences do not cluster as straightforwardly. The tunicate and lancelet sequences are relatively similar to each other, but well separated from the rest of the sequences. Although the exons are highly similar, Kappé et al. have reported that the intron placement is quite different [129]. In the *Ciona intestinalis* protein, as in vertebrate γ-crystallins, the introns are located between the regions coding for the Greek key motifs, whereas the introns are found within these motifs in the lancelet protein [129]. The latter also lacks the short N-terminal extension encoded by a separate exon in the tunicate protein and vertebrate γS-crystallins. The hagfish and lamprey crystallins are well separated from each other and from the next major cluster, which contains the γS-crystallins. The sequences found in Cluster 3 of the protein tree are found in three subclusters, containing γM-crystallins from the zebrafish and coelacanth, the mammalian and alligator γD-crystallins, and γ-crystallins from the African clawed frog and slender bamboo shark. Despite their annotations as γS1, γS2, and γM1, all three shark crystallins are highly similar to each other, and all have a methionine content of 9–10%. The presence of these three subclusters, as well as the groupings of crystallins from highly divergent organisms (e.g., zebrafish and coelacanth, frog and shark) and the inclusion of the alligator sequence with the mammalian γD-crystallins may suggest that γ-crystallin gene loss has commonly occurred during crystallin evolution. Finally, the rest of the γM-crystallin sequences are related by a highly chained structure.

The most striking observation from the phylogenetic clustering is that Cys positions seem to be a stronger predictor of which γ-crystallin partitions to which cluster than even the proteins’ current classification in the proteomic databases. Crystallins containing the “DCDCDC” loop (or its close homologs) found in human γS-crystallin clustered together in Cluster 2. By contrast, even proteins that are currently classified as γS-crystallins but that lack this signature sequence partitioned to Cluster 3 along with γD-crystallins, as shown in Appendix A. Thus, even though there is very low overall conservation of Cys within the βγ-crystallin superfamily, specific Cys-containing motifs appear to be linked to functional specialization within the γ-crystallin family. This not only reveals somewhat surprising evolutionary relationships but suggests more appropriate annotations for many of these proteins. For example, the clustering of fish and frog crystallins with mammalian γD-crystallins suggests that γD-like crystallins are more common and widespread than was previously supposed. Furthermore, the positioning of the fish γM-crystallins in the nucleic acid tree indicates that they may have been derived from γD-like crystallins rather than γS-crystallins.

### 3.3. Structural Analysis of Crystallin Homologs

The βγ-crystallin fold is an ancient motif that first arose in microbial metal ion-binding proteins [130,131]. Each Greek key consists of four interleaved β-strands (Figure 3). In order to explore the structural significance of high Cys content in γ-crystallins using our phylogenetically representative set of γ-crystallins, we downloaded experimentally determined structures from the Protein Data Bank for cases where they are available; for the other cases, structures were predicted using D-I-TASSER, the current most accurate machine-learning algorithm for protein structure prediction from multiple sequence alignments [61].

Protein structure visualizations were generated using ChimeraX [135,136]. The lowest-energy predicted structures of all the βγ-crystallins examined here show the same overall fold (models provided in the online Appendix A). Although ion binding dramatically stabilizes tunicate βγ-crystallin, the two Ca^2+^ ions are coordinated to the loops internal to each Greek key, where they do not perturb the β-strand structure. Multiple sequence alignment of a large number of related sequences using DeepMSA (Appendix A) showed that the strongly conserved residues—those most likely to be critical for maintaining the fold—are Ser and Gly, along with the tryptophans commonly found in the hydrophobic core of each Greek key domain, the tyrosines associated with the tyrosine corners, and some surface-exposed charged residues.

Although they are not strongly conserved among crystallins generally, Cys and Met are highly important for lens crystallin function. The Cys and Met contents of the crystallins investigated here are given in Table 1 and Appendix A. Initiator methionines are assumed to be cut off in the mature form of eukaryotic proteins and are therefore not included in the total. The positioning of cysteines in solution NMR structures for representative γ-crystallins is shown in Figure 4. Human γS-crystallin, human γD-crystallin, and zebrafish γM-crystallin were chosen because of the availability of solution NMR structures for these three proteins. The top row shows human γS-crystallin: Panel A shows the protein surface, revealing that all three cysteines in the N-terminal domain are at least partially exposed. The two cysteines in the C-terminal domain are fully buried. Panel B shows a ribbon diagram with all of the cysteines as ball and stick models. Their sequence positions are labeled; C25 is the most prominently exposed to solvent, although C23 and C27 are partially exposed as well. Panel C shows a view of the N-terminal Cys loop with the protein rotated 90° relative to the other views. The middle row shows human γD-crystallin. Panel A shows the protein surface. Only C19 (C18 in the PDB numbering scheme of Basak et al. [51]) is present in a position corresponding to the Cys loop of γS; it is fully buried. In contrast, the two C-terminal cysteines are partially exposed. Panel B shows a ribbon diagram with the cysteine residues labeled. Panel C shows a rotated view of the N-terminal Cys. The bottom row (Panels G–I) shows zebrafish γM7-crystallin. Like human γD-crystallin, it has only one Cys residue (C19) in the solvent loop; however, it is partly solvent exposed while its C-terminal cysteines are buried.

The positions of Met residues for the same proteins are shown in Figure 5. Human γS (Panel A) and γD-crystallin (Panel B) have five and four methionines, respectively, representing an abundance of about 5% in each case. In contrast, zebrafish γM7-crystallin (Panel C) has 16 Met residues or 9.2%, a value that is high compared to typical proteins, but on the low side for fish γM-crystallins, which can have a Met content of 20% or higher. The highest Met content for the proteins examined here is 22.5% for γM1-crystallin from *Chiloscyllium indicum* (slender bamboo shark).

As discussed in Section 2, high Cys content puts lens γ-crystallins at a high risk of disulfide-driven aggregation, though it also allows them to function as oxidoreductases. Using our set of solution NMR and predicted D-I-TASSER structures, we therefore analyzed solvent accessibility of Cys residues. Buried Cys is less likely to form disulfides, unless or until the protein adopts a misfolded conformation for a sufficiently long time. By contrast, natively solvent-exposed Cys residues present a more immediate risk: if there are solvent-exposed Cys residues in at least two distinct regions of the protein’s surface, then head-to-tail aggregates may form via intermolecular disulfides. By contrast, if there is only one solvent-exposed Cys residue or residue cluster on the protein’s surface, then aggregation will stop at the head-to-head dimer stage.

Our analysis revealed a striking observation, shown in Figure 6; the γ-crystallins in our set had solvent-exposed Cys residues in the N-terminal domain (NTD) or in the C-terminal domain (CTD), but not both. The only exception was γM2b-crystallin from zebrafish. Of course, domain-level Cys accessibility analysis is still a relatively crude metric. The data overall strongly suggest that avoidance of multiple intermolecular disulfides per molecule, given the ensuing light-scattering aggregation, has been a significant evolutionary constraint of the γ-crystallin family.

## 4. Discussion

### 4.1. Evolutionary Pressure for Low Light Scattering by Lens γ-Crystallins over a Lifetime

For our discussion of γ-crystallins, we focus on the three most common types in extant vertebrates: γS-crystallins, γD-crystallins, and γM-crystallins. In humans, γD-crystallin is concentrated primarily in the lens core, whereas γS-crystallin is primarily in the lens cortex, where protein expression continues throughout life [138]. Based on our analyses of the sequences and predicted structures of these proteins, we propose that the most easily identifiable difference between γS- and γD-like crystallins, which have traditionally been differentiated based on features such as the hydrophobicity of the interdomain interface [66], is whether the DCDCD motif is present in the N-terminal domain. In humans, γS- and γD-crystallins are at the extreme ends of the spectrum in terms of NTD and CTD Cys exposure, respectively (Figure 6), and γS-crystallins in general have at least two of these DC pairs (Appendix A).

In other vertebrates, the number of γ-crystallin paralogs and their distribution varies considerably. Teleost fish have many paralogs of γM-crystallins, which share the same double-Greek key fold with γS and γD but are distinguished by their unusually high methionine content. Unlike the lenses of terrestrial organisms, which derive some optical power from an air–water interface at the cornea, the crystallins in fish lenses must provide all the refraction needed to focus an image on the retina. The Met-rich γM-crystallins are thought to contribute to this function due to the very high refractivity of the Met side chain. The multitude of paralogs may also aid in maintaining solubility at the extreme concentrations found in the fish lens, much in the same way that frustration prevents crystallization in an ionic liquid. Another hypothesis that has been proposed is that the large number of surface-exposed Met residues is responsible for mediating solvent interactions, making γM-crystallins the least strongly hydrated of the structural crystallins, a feature that may aid in packing them in the crowded fish lens [139].

### 4.2. Evolutionary Pressure for High Optical Power of Lens γ-Crystallins

Refractive index depends on lens shape, hydration, and protein structure, as well as post-translational modifications [99,140]. The lens has a gradient of both protein concentration and distribution, with the highest protein concentration overall as well as the highest abundance of γ-crystallins near the central nucleus of the lens [141]. The optical properties of the lens are also influenced by the refractivity of the proteins themselves. The traditional understanding of protein refractivity is based on the additive Gladstone–Dale model, where the refractivity of a protein is determined by the weighted average of the refractivities of its component amino acids [19,142,143]. This model does not account for the differences in surface hydration between a collection of separate amino acids and a folded protein, nor potential interactions between pairs of aromatic residues, which are very highly enriched in lens crystallins relative to other proteins. Experimental measurements have shown that for lens crystallins, the Gladstone–Dale model significantly underestimates the refractivity based on sequence alone; a simple correction accounting for interactions between pairs of aromatic residues can correct for about two-thirds of this error [144]. Aside from structural factors, analysis of crystallin sequences reveals that they have very unusual amino acid compositions. They are highly enriched in the most polarizable residues (e.g., Arg, Cys, Met, Phe, Tyr, and Trp), at the expense of aliphatic residues that contribute little to refractivity.

The dual selection of crystallins for refractivity and solubility leads to some inherent tension, as the residues that best promote solubility (e.g., small polar and/or charged residues) are not the ones that most increase refractivity. The model of W. D. Hamilton [55] predicts that improved refractive power in younger age should be selected by evolution even at the cost of comparable increase in light scattering due to aggregation in old age—and that this should hold true even in fish or other species whose reproductive fitness increases with age.

Both cysteine and methionine significantly enhance refraction in the lens, a phenomenon that is best illustrated by the unusually high content of both, but especially methionine, in the fish γM-crystallins. In fish, the lens provides all the focusing power of the eye, so the evolutionary pressure for the high refractive index increment is especially intense; accordingly, fish crystallins are packed to concentrations of up to 1000 mg/mL and are highly refracting [23]. For the γ-crystallins found in Table 1, the average Cys content is 5.6% and the Met content is 10.9% for γM-crystallins, compared to 3.6% and 2.4% for βγ-crystallins, 3.9% and 2.2% for γS-crystallins, and 1.7% and 2% for eukaryotic proteins in general [2]. Each of their side chains has a large, polarizable sulfur atom that contributes greatly to refractivity; however, these moieties are also susceptible to oxidation and chemical damage that may compromise solubility.

In particular, Cys plays many roles in proteins, such as functional redox chemistry and stabilizing disulfide bonds, but it can also powerfully promote misfolding and aggregation if incorrect intra- or intermolecular disulfides are formed. The only way to avoid Cys while retaining the high polarizability of sulfur is to replace it with Met; however, Met is even more readily oxidized than Cys, and the resulting sulfoxides could disrupt the core packing and/or the local hydration environment of the protein. Comparison of the predicted Gladstone–Dale refractive index increment (the change in refractive index per concentration increment, dn/dc) for lens crystallins to other proteins with Greek key motifs suggests that the lens crystallins are more refractive than others with a similar secondary structure, even though other Greek key proteins are also highly stable [145]. This hints at the tension between the dual functions of lens crystallins; amino acids that are selected for stability and solubility do not necessarily provide high refractivity and vice versa. These authors also found that the K/R ratio in βγ-crystallins was lower than for comparable proteins, which they attribute to selection for stabilizing salt bridges. Alternatively, it may be due to selection for Arg itself, which has a higher reactive index increment than Lys. In fact, substituting Arg (dn/dc = 0.206 mL/g) for Lys (dn/dc = 0.181 mL/g) represents a major increase in refractivity without changing the charge of the residue.

Our examination of the abundance and positioning of cysteine and methionine in vertebrate γ-crystallins and their relatives in invertebrate chordates suggests that these residues provide the key to understanding the differences among types of γ-crystallins and generating hypotheses about their functionality in the lens. Based on the requirement for higher refractivity in the aquatic lens than its terrestrial counterpart, we would expect crystallins from aquatic organisms to be more enriched in Cys and Met than their terrestrial homologs. This is most dramatically exemplified by the large enrichment in these amino acids in the fish-specific γM-crystallins, but it can also be observed for other aquatic crystallins in this data set. Notably, among the β-crystallins, which are assumed to have low refractivity in mammals, many fish sequences are highly enriched in Cys and Met even when their terrestrial homologs are not. For example, several of the βB1-crystallins from both *Danio rerio* and *Latimeria chalumnae* have Met contents in the 7% range, compared to 2% for mouse and 1.5% for human. This is notable because the zebrafish (ray-finned fish) and coelacanth (lobe-finned fish) are not close relatives, suggesting that the similarity is due to convergence in function. The increased abundance of Cys and Met concentration in the lenses of aquatic organisms (and those that require good vision, as opposed to e.g., the tunicate) may be selected due to increased refractivity, altered hydration properties, or both. These hypotheses require experimental validation, some of which is underway in our own and other labs.

### 4.3. Synergies between Sulfur-Containing and Aromatic Residues in Lens γ-Crystallins

Much of our discussion above highlights the evolutionary tension between the requirements for high optical power and high long-term transparency of the lens. We have speculated that these competing evolutionary pressures may have pushed γ-crystallins and βB-crystallins toward opposite extremes of Cys (and Met) abundance or depletion. Yet, these two broad phenotypic requirements do not always or necessarily conflict. Sulfur–aromatic interactions may contribute to the thermodynamic stability of these proteins [146,147], but they may also contribute to anomalously high refractive index increments that are beyond what can be calculated from a primary structure alone, as is the case for pi-pi and cation-pi interactions. Accordingly, convergent evolution has favored high sulfur and aromatic content not only in crystallins but also in reflectins, another protein family with an exceptionally high refractive index [148]. Additionally, sulfur atoms may protect aromatic residues from irreversible damage by channeling away high-energy excitons produced by UV irradiation [47,149], while the aromatics, by absorbing those photons and channeling them to the Cys residues, in particular, could photochemically reduce disulfide bonds [32,150,151]. Strategically positioned reversible disulfides might not only turn crystallins into a proteinaceous redox buffer but also render the crystallin domains “self-healing”, capable of resisting the deleterious effects of certain denaturing stresses.

## 5. Conclusions

Much remains to be learned about how evolution has harnessed and balanced the many unique properties of sulfur-containing amino acids to optimize and preserve vision. We have reviewed available evidence from the experimental literature, genomic datasets, and ML-based protein structure prediction. We have found support for our hypotheses that lens crystallin families may have evolved to solve a trade-off between high refractive power and low late-life aggregation propensity and that the evolutionary record contains a signature of optimized spatial positioning of Cys residues to maximize useful redox activity while minimizing deleterious light-scattering aggregation. In addition, we found reason to suggest re-evaluating the evolutionary relationships among γ-crystallin subfamilies, as the γM and γD crystallin subfamilies in our dataset are unexpectedly similar with respect to their evolutionarily tuned Cys residue distributions. Based on our present review and investigation, we may hazard several experimentally testable hypotheses to focus future research in this area. Do crystallin thermodynamic and kinetic parameters vary predictably with an organism’s lifespan? Have crystallins in aquatic vertebrates faced a different balance of selective pressures for refractivity and transparency from those of land-dwelling ones? Do sulfur–aromatic interactions couple redox chemistry and photochemistry in the eye lens? We highlight the need for systematic comparisons of the biophysical and biochemical properties of lens crystallins across species and across crystallin subfamilies as one way of testing these hypotheses. We have previously offered conceptual models of disulfide flow and redox buffering in lens γ-crystallins [32,38,85]. Improved empirical understanding of the origins, benefits, and drawbacks of high Cys content in the γ-crystallin family will improve future therapeutic strategies to postpone or prevent the progression of turbidity in the aging lens.

## Figures and Tables

**Figure 1 biomolecules-14-00594-f001:**
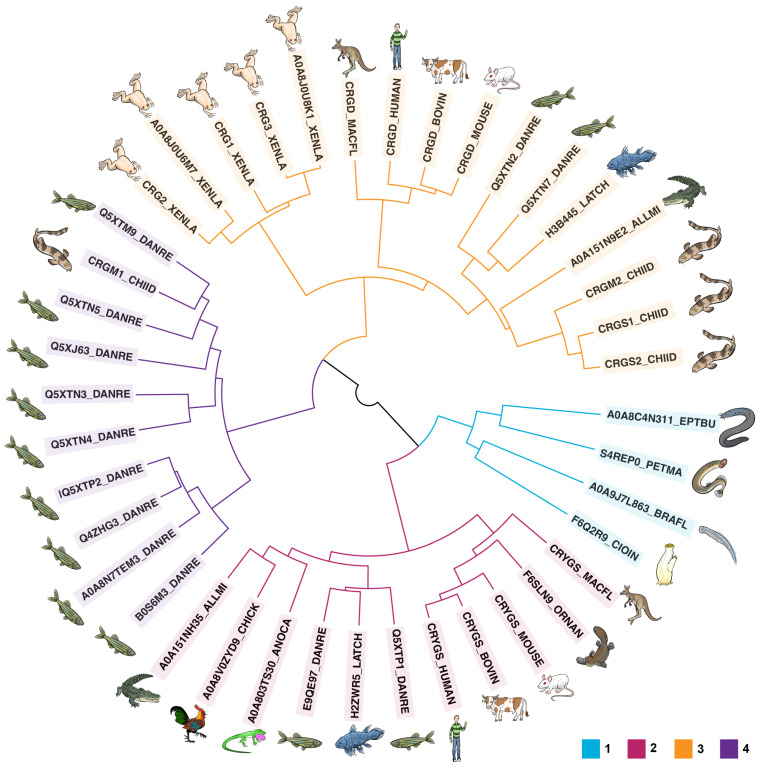
Protein sequence similarity among βγ- and γ-crystallins. Crystallin sequences from chordates (tunicate, lancelet, hagfish) cluster with that of the sea lamprey, a representative of the most basal vertebrates (Cluster 1). The next major group of crystallins to split (Cluster 2) include γS-crystallins from many organisms. From there, two more large groups of γ-crystallins split off. Group 3 includes mammalian γD-crystallins (kangaroo, human, cow, mouse). as well as other γ-crystallins found in lobe-finned fish (coelacanth), chondrichthyans (bamboo shark), bony fish (zebrafish), crocodilians (alligator), and amphibians (clawed frog). Finally, group 4 comprises fish-specific γM-crystallins from the zebrafish and bamboo shark.

**Figure 2 biomolecules-14-00594-f002:**
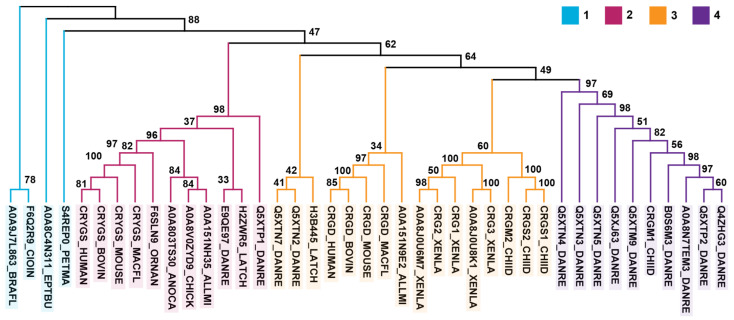
Phylogenetic tree for βγ- and γ-crystallins. Nucleic acid sequences coding for crystallin proteins from the tunicate and lancelet are highly similar. The labels are color coded according to the protein sequence clusters described in Figure 1.

**Figure 3 biomolecules-14-00594-f003:**
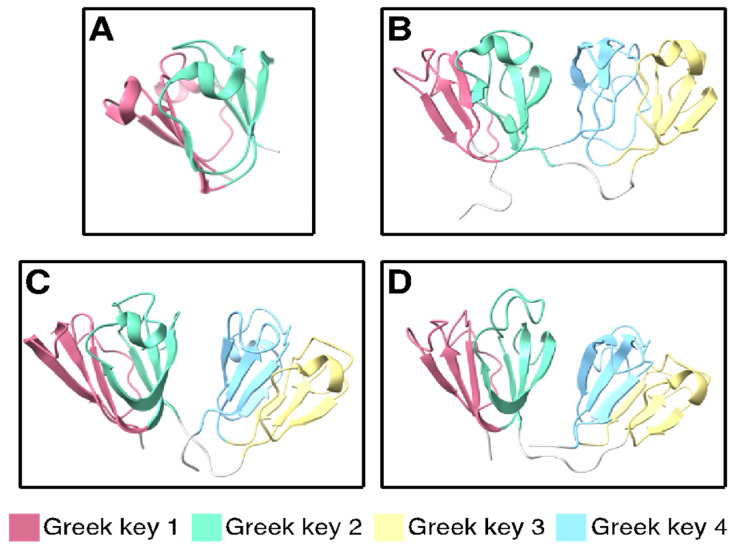
Experimentally determined structures of representative βγ- and γ-crystallins. The double Greek key fold is common to βγ-crystallins. (**A**) Tunicate (*Ciona intestinalis*) βγ-crystallin (PDB ID: 2BV2) [132]. (**B**) Human γS-crystallin (PDB ID: 2M3T) [58]. (**C**) Human γD-crystallin (PDB ID: 2KLJ) [133]. (**D**) Zebrafish γM7-crystallin (PDB ID: 2M3C) [134]. The individual Greek key motifs are colored in pink, green, yellow, and blue from N- to C-terminus.

**Figure 4 biomolecules-14-00594-f004:**
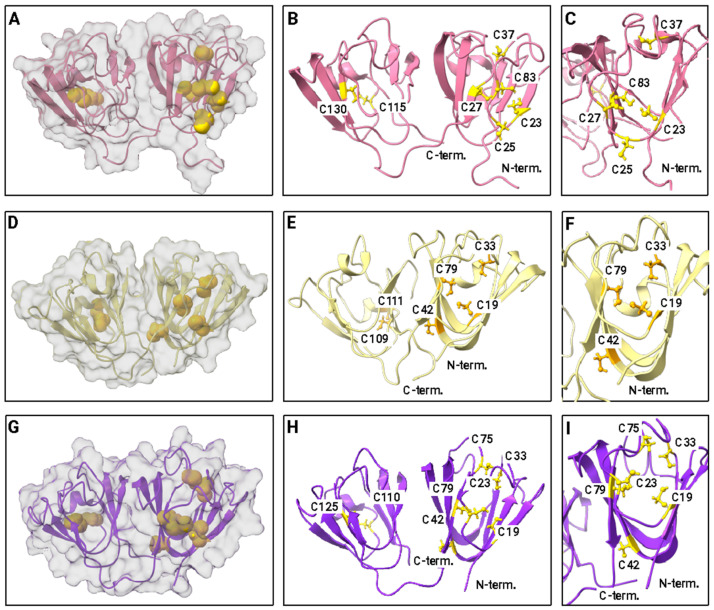
Cysteine in representative γ-crystallins. (**A**–**C**) Human γS-crystallin (PDB ID: 2M3T) [58]. Inset of the cysteine loop, formed by C23, C25, and C27 along with surrounding residues. (**D**–**F**) Human γD-crystallin (PDB ID: 2KLJ) [133]. (**G**–**I**) Zebrafish γM7-crystallin (PDB ID: 2M3C) [134]. Note that we use the UniProt convention for numbering γ-crystallin residues in this paper, counting Met1 as the first residue even though it is absent from the mature form of the protein. In other papers, we and others have used the PDB numbering scheme of Basak et al. [137] or the traditional scheme based on alignment to human γB-crystallin.

**Figure 5 biomolecules-14-00594-f005:**
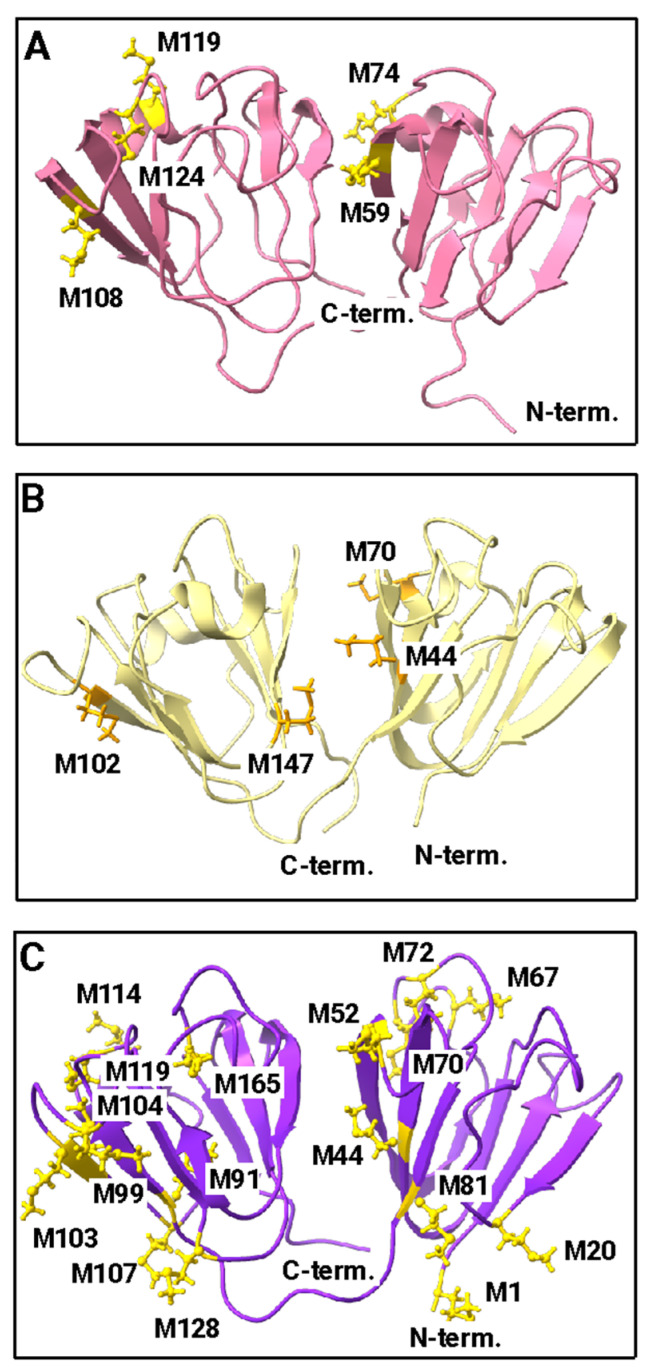
Methionine in representative γ-crystallins. (**A**) Human γS-crystallin (PDB ID: 2M3T) [58]. (**B**) Human γD-crystallin (PDB ID: 2KLJ) [133]. (**C**) Zebrafish γM7-crystallin (PDB ID: 2M3C) [134].

**Figure 6 biomolecules-14-00594-f006:**
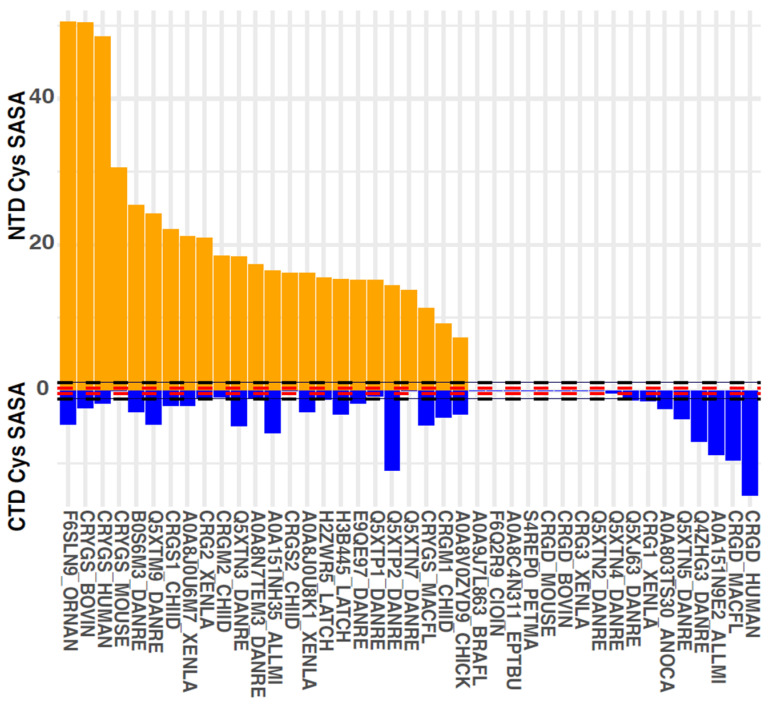
Combined solvent-accessible surface area (SASA), in square Angstroms, of Cys residues by domain in solution NMR structures or D-I-TASSER predicted structures of representative γ-crystallins. Protein nomenclature follows UniProt. Orange = N-terminal cysteines; blue = C-terminal cysteines. Red dashed lines indicate the mean SASA for a control buried Cys that is conserved in almost all of the γ-crystallins, e.g., C83 in human γS (see Figure 4C). Black dashed lines indicate the mean + standard deviation for the control buried SASA.

**Table 1 biomolecules-14-00594-t001:** Cysteine and methionine content of βγ- and γ-crystallins in representative chordates.

Organism	Protein	UniProt ID	Method	Length	Cys (no./%)	Met (no./%)
*Branchiostoma floridae*	S-crystallin	C3YKG6_BRAFL	1	128	6/4.7	4/3.1
*Ciona intestinalis*	Βγ-crystallin	F6Q2R9_CIOIN	3	84	0/0	0/0
*Eptatretus burgeri*	Βγ-crystallin	A0A8C4N311_EPTBU	3	153	7/4.6	3/2.0
*Petromyzon marinus*	γS-crystallin	S4REP0_PETMA	3	176	9/5.1	8/4.5
*Danio rerio*	γS1-crystallin	E9QE97_DANRE	1	178	8/4.5	2/1.1
*Danio rerio*	γS2-crystallin	Q5XTP1_DANRE	1	174	8/4.6	2/1.1
*Latimeria chalumnae*	γS-crystallin	H2ZWR5_LATCH	3	178	7/3.9	6/3.4
*Ornithorhynchus anatinus*	γS-crystallin	F6SLN9_ORNAN	1	178	6/3.4	3/1.7
*Macropus fuliginosus*	γS-crystallin	CRYGS_MACFL	4	178	8/4.5	4/2.2
*Mus musculus*	γS-crystallin	CRYGS_MOUSE	1	178	7/3.9	4/2.2
*Bos taurus*	γS-crystallin	CRYGS_BOVIN	1	178	6/3.4	6/3.4
*Homo sapiens*	γS-crystallin	CRYGS_HUMAN	1	178	7/3.9	5/2.8
*Anolis carolinensis*	γS-crystallin	A0A803TS30_ANOCA	1	179	5/2.8	5/2.8
*Alligator mississippiensis*	γS-crystallin	A0A151NH35_ALLMI	2	182	7/3.8	4/2.2
*Gallus gallus*	γS-crystallin	A0A8V0ZYD9_CHICK	1	175	7/4.0	3/1.7
*Macropus fuliginosis*	γD-crystallin	CRGD_MACFL	4	174	8/4.6	8/4.6
*Mus musculus*	γD-crystallin	CRGD_MOUSE	1	174	7/4.0	7/4.0
*Bos taurus*	γD-crystallin	CRGD_BOVIN	4	174	5/2.9	5/2.9
*Homo sapiens*	γD-crystallin	CRGD_HUMAN	1	174	6/3.4	5/2.9
*Alligator mississippiensis*	γD-crystallin	A0A151N9E2_ALLMI	2	175	6/3.4	7/4.0
*Xenopus laevis*	γ-crystallin 1	CRG1_XENLA	2	175	5/2.9	5/2.9
*Xenopus laevis*	γ-crystallin 2	CRG2_XENLA	4	175	7/4.0	6/3.4
*Xenopus laevis*	γ-crystallin 3	CRG3_XENLA	2	175	6/2.4	5/2.9
*Xenopus laevis*	γ-crystallin 4	A0A8J0U6M7_XENLA	1	175	7/4.0	6/3.4
*Xenopus laevis*	γ-crystallin 5	A0A8J0U8K1_XENLA	1	175	6/3.4	5/2.9
*Chiloscyllium indicum*	γS1-crystallin	CRGS1_CHIID	4	173	7/4.0	10/5.8
*Chiloscyllium indicum*	γS2-crystallin	CRGS2_CHIID	2	173	6/3.5	9/5.2
*Chiloscyllium indicum*	γM2-crystallin	CRGM2_CHIID	4	176	8/4.5	9/5.1
*Latimeria chalumnae*	γM2-crystallin	H3B445_LATCH	3	176	9/5.1	4/2.3
*Danio rerio*	γS3-crystallin	Q5XTN7_DANRE	1	183	6/3.3	7/3.8
*Danio rerio*	γS4-crystallin	Q5XTN2_DANRE	1	176	7/4.0	2/1.1
*Chiloscyllium indicum*	γM1-crystallin	CRGM1_CHIID	4	120	10/8.3	27/22.5
*Danio rerio*	γM1-crystallin	Q5XTN6_DANRE	1	178	8/4.5	19/10.7
*Danio rerio*	γM2-crystallin	A0A8N7TEM3_DANRE	1	174	10/5.7	22/12.6
*Danio rerio*	γM2a-crystallin	Q4ZHG3_DANRE	1	181	10/5.5	24/13.3
*Danio rerio*	γM2c-crystallin	Q5XTP2_DANRE	1	175	10/5.7	22/12.6
*Danio rerio*	γM2d1-crystallin	B0S6M3_DANRE	2	175	10/5.7	23/13.1
*Danio rerio*	γM3-crystallin	Q5XTM9_DANRE	1	174	10/5.7	13/7.5
*Danio rerio*	γM4-crystallin	Q5XTN5_DANRE	1	174	9/5.2	10/5.7
*Danio rerio*	γM5-crystallin	Q5XJ63_DANRE	1	177	11/6.2	12/6.8
*Danio rerio*	γM6-crystallin	Q5XTN4_DANRE	2	177	7/4.0	11/6.2
*Danio rerio*	γM7-crystallin	Q5XTN3_DANRE	1	174	8/4.6	16/9.2

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
