# Peer review of "The Functional Significance of High Cysteine Content in Eye Lens γ-Crystallins"

_biomolecules, 2024, doi:10.3390/biom14050594_

Round 1

Reviewer 1 Report

Comments and Suggestions for Authors

I applaud Eugene for his exceptional work on this critical topic. The manuscript is impeccably written and easily comprehensible. I do have one suggestion for the authors to consider. Given the incorporation of evolution into this discussion, it may be beneficial to include discussions about the current widely accepted theory of aging evolutionary mechanisms, as initially proposed by WD Hamilton in 1966. This theory posits that due to ineffective selection later in life, deleterious mutations tend to accumulate. Consequently, mutations that are advantageous early in life may become detrimental later on due to this ineffective selection pressure. Could the enrichment of cysteine residues in gamma crystallins contribute to this diminished selection pressure?

Reviewer 2 Report

Comments and Suggestions for Authors

This work represents a good overview of information accumulated by humanity on beta- and gamma-crystallins, raising interesting questions about the role of Cys restudies within the lens protein. To answer the raised questions, the authors suggest a few attempts to analyse the accumulated data using modern bioinformatics methods. This analysis focuses on testing three hypotheses on the role of Cys and Met in the evolution-derived stabilisation of beta and gamma-crystallins.

In my opinion, the analysis of the Cys residue conservation in crystallins of different species and the correlation of residue content with refractivity sound very reasonable, however, as the authors correctly stated, should be further confirmed experimentally. The phylogenetic clustering of gamma-crystallins is a good attempt to find correlations between species, however, probably due to the limits of the available experimental data, this analysis does not provide any valuable conclusions except suggestions for more correct annotation of proteins. In the case of disagreement with the last comment, the importance of this analysis could be additionally highlighted by authors for a Reader's convenience.

In conclusion, I suggest this work for the publication after corrections of minor remarks:

1.       Section 3.3 should be revised according to the following remarks:

A.      Figures 1-3, analysed only in this section, should be moved closer to text describing them for a Reader’s convenience. The enumeration of Figures should be changed accordingly.

B.      Strings 418-423. Sentences repeat the information stated several times throughout the review and could be safely removed for manuscript clarity.

C.      Strings 424-450. The amino acids are mentioned in their full names, while three-letter naming was used in the main part of the manuscript. The AA naming should be unified.

2.       String 478 and Figure 6. NTD and CTD abbreviatures should be disclosed at first mention in the text.

3.       String 541. The ratio dn/dc also should be disclosed at the first mention.

4.       Table 1. The caption is missing.

5.       Reference 146 should be updated as recently published work in a peer-reviewed journal.

Reviewer 3 Report

Comments and Suggestions for Authors

The manuscript entitled The functional significance of high cysteine content in eye lens γ-crystallins brings valuable information about the mechanism of crystallyn opacifiation and how oxidative stress may contribute to this mechanism.

Observation

Abstract

What do you mean by cystein residues?

Beta -crystallin family and gamma cristallyns subfamily ? Since they do no explain these family, is difficult to understand the context.

The abstract is difficult to understand due to using to many subclasses of crystalins structures , without explanation of these components.

I suggest to make the abstract more general to be easy understanding by the readers.

Introduction 

They should explain the organisation and the functions of crystallin families. There are two main crystallin gene families: alpha-crystallins, and betagamma-crystallins. alpha-crystallins are molecular chaperones that prevent aberrant protein interactions. The chaperone properties of alpha-crystallin are thought to allow the lens to tolerate aging-induced deterioration of the lens proteins without showing signs of cataracts until older age. alpha-crystallins not only possess chaperone-like activity in vitro, but can also remodel and protect the cytoskeleton, inhibit apoptosis, and enhance the resistance of cells to stress. Recent advances in the field of structure-function relationships of alpha-crystallins have provided the first clues to their underlying roles in tissues outside the lens. Proteins of the betagamma-crystallin family have been suggested to affect lens development, and are also expressed in tissues outside the lens (doi: 10.1016/j.preteyeres.2006.10.003.)

They also should explain more mechanisms of lens opacification in Introduction, including the diseases that could contribute to that.

- line 32 - what do you mean by "canonical"? You should keep the academical sentences in your manuscript.

- line 92-94 - you need to answer to your own question at least by a supposition or by a theoretical assumption.

- line 98 - please mention the scientific name of Tunicate and replace it everywhere into the text.

What is the reason you compared the human cristallyns with Zebra fish and Tunicates crystallyns? Your title should contain these comparative aspects because you refer to them in a large part of your manuscript.

- line 122 - what do you mean by this sentence? Do you refer to life expectancy improving? Than you need to bring more arguments to this idea than fitness. Moreover the excessive fitness increase the oxidative stress in the body that could interfere with lens opacification.

- line 173 - this hypothesis has not enough sustainability, it is only a hazardous presumption.

- line 203-204 - indicate the references here.

- line 195 - besides age onset cataract there are many other mechanisms for lens opacity acceleration. Please give some examples, because age related cataract is only a segment from all disorders that can initiate the lens opacification.

Table pg 17 - please insert this table into the text.

Please make two different chapters with Discussions and Conclusions, to understand why your manuscript is bringing novelty in the field of lens opacification theories. 

Reviewer 4 Report

Comments and Suggestions for Authors

The authors attempted to answer this curious question in that the Cys content of γ-crystallins was above the average for human proteins and took advantage of expanding genomic databases and improved machine learning tools for protein structure prediction to investigate it further. The purpose of this study is quite interesting. However, several points and questions need to be explained and answered. 

This is well well-written review article regarding the Cys content of γ-crystallins. However, several viewpoints should be added in their review to increase the reader’s interest. First, long-term exposure to ultraviolet rays is one of the well-known risk factors to aggravate the cataract. The relationship between UV exposure and Cys content of γ-crystallins can be added. Second, the ongoing therapeutic approach to manipulate the Cys content of γ-crystallins to delay cataractogenesis can also be reviewed. 

Round 2

Reviewer 4 Report

Comments and Suggestions for Authors

The authors did their best to revise the manuscript as the reviewers' advice. 

Comments on the Quality of English Language

The quality of English language was fine.